# Architectural principles for Hfq/Crc-mediated regulation of gene expression

Xue Yuan Pei[1†], Tom Dendooven[1†], Elisabeth Sonnleitner[2], Shaoxia Chen[3], Udo Bläsi[2]*, Ben F Luisi[1]*

[1]Department of Biochemistry, University of Cambridge, Cambridge, United Kingdom; [2]Department of Microbiology, Immunobiology and Genetics, Max F Perutz Laboratories, Center of Molecular Biology, University of Vienna, Vienna Biocenter, Vienna, Austria; [3]MRC Laboratory of Molecular Biology, Cambridge, United Kingdom

**Abstract** In diverse bacterial species, the global regulator Hfq contributes to post-transcriptional networks that control expression of numerous genes. Hfq of the opportunistic pathogen *Pseudomonas aeruginosa* inhibits translation of target transcripts by forming a regulatory complex with the catabolite repression protein Crc. This repressive complex acts as part of an intricate mechanism of preferred nutrient utilisation. We describe high-resolution cryo-EM structures of the assembly of Hfq and Crc bound to the translation initiation site of a target mRNA. The core of the assembly is formed through interactions of two cognate RNAs, two Hfq hexamers and a Crc pair. Additional Crc protomers are recruited to the core to generate higher-order assemblies with demonstrated regulatory activity in vivo. This study reveals how Hfq cooperates with a partner protein to regulate translation, and provides a structural basis for an RNA code that guides global regulators to interact cooperatively and regulate different RNA targets.
DOI: https://doi.org/10.7554/eLife.43158.001

*For correspondence:
udo.blaesi@univie.ac.at (UBä);
bfl20@cam.ac.uk (BFL)

†These authors contributed equally to this work

Competing interests: The authors declare that no competing interests exist.

## Introduction

The RNA chaperone Hfq contributes to the control of mRNA translation through different modes of action. In one mode, Hfq acts indirectly by facilitating base-pairing interactions of cognate mRNA targets with small regulatory RNAs (*Vogel and Luisi, 2011*; *Wagner and Romby, 2015*). In a second mode, the RNA chaperone can directly repress translation independently of small RNAs, by binding A-rich sequences at or in the vicinity of translation initiation sites (*Vecerek et al., 2005*; *Salvail et al., 2013*; *Sonnleitner and Bläsi, 2014*). As a member of the extensive Lsm/Sm protein family, Hfq shares an ancient structural core that oligomerizes to form toroidal architectures exposing several RNA-binding surfaces. Crystallographic and biophysical data showed that RNA recognition is mediated by distinct interactions with distal, proximal and rim faces of the hexameric ring (*Schumacher et al., 2002*; *Link et al., 2009*; *Sauer et al., 2012*; *Horstmann et al., 2012*; *Panja et al., 2013*), as well as revealing the role of the unstructured C-terminal tail in auto-regulating RNA-binding activities (*Santiago-Frangos et al., 2016*; *Santiago-Frangos et al., 2017*).

In the opportunistic, Gram-negative pathogen *Pseudomonas aeruginosa*, Hfq acts as a pleiotropic regulator of metabolism (*Sonnleitner and Bläsi, 2014*), virulence (*Sonnleitner et al., 2003*; *Fernández et al., 2016*; *Pusic et al., 2016*), quorum sensing (*Sonnleitner et al., 2006*; *Yang et al., 2015*) and stress responses (*Lu et al., 2016*). Many of these roles are likely facilitated through partner molecules, and numerous putative protein interactors of *P. aeruginosa* Hfq have been identified with functions in transcription, translation and mRNA decay (*Van den Bossche et al., 2014*). In the case of Hfq from *E. coli*, the functionally important partners include RNA polymerase, ribosomal protein S1 (*Sukhodolets and Garges, 2003*), the endoribonuclease RNase E (*Ikeda et al., 2011*),

**eLife digest** Living things can adapt rapidly to changes in their surroundings by switching whole groups of genes on and off. These responses must be controlled carefully, and they are often coordinated by regulatory proteins working together. Within a cell, the coded information in genes is copied to create molecules called mRNAs, which are then translated to produce proteins. Stopping the cell from reading the information in mRNAs is one way of shutting down specific genes.

*Pseudomonas aeruginosa* is a species of bacteria that can infect humans and can cause cases of sepsis and pneumonia. It changes the activity of its genes in response to its environment. For example, certain genes are only active inside a human host. This allows the bacteria to make the best use of available nutrients and energy. In these cells, two proteins named Hfq and Crc cooperate to silence groups of genes. They do this by stopping the cell from reading specific mRNA molecules, but how they do this is not fully understood.

By using a technique called cryo-electron microscopy, Pei, Dendooven et al. studied Hfq and Crc attached to mRNAs. The results show that two groups of six Hfq molecules and two Crcs attach themselves to two mRNA sections to create a structure that stops an mRNA from being translated into a protein. Since the structure only forms with certain mRNAs, the effect is specific to certain genes. The structure needs both Hfq and Crc to work together, which means it only forms in specific situations – when the affected genes are not needed.

*P. aeruginosa* is highly antibiotic resistant and new drugs are urgently needed to control infections. Understanding how this disease-causing bacterium controls its genes could lead to new treatments. The mechanisms of gene regulation are also common to many other forms of life so may also aid the wider understanding of how cells adapt to rapidly changing environments.

DOI: https://doi.org/10.7554/eLife.43158.002

polyA-polymerase, and the exoribonuclease polynucleotide phosphorylase (*Mohanty et al., 2004*; *Bandyra et al., 2016*). Most likely, these complexes are RNA mediated and affect the co-localisation of the machineries of transcription, translation and RNA decay (*Worrall et al., 2008*; *Resch et al., 2010*; *Vecerek et al., 2010*).

One *P. aeruginosa* protein that was found to co-purify with tagged Hfq is the Catabolite repression control protein, Crc (*Van den Bossche et al., 2014*; *Moreno et al., 2015*; *Sonnleitner et al., 2018*). Crc is involved in carbon catabolite repression (CCR) in *Pseudomonas*, a process that channels metabolism to use preferred carbon sources (such as succinate) until they are exhausted, whereupon alternative sources are used (*Rojo, 2010*). In addition to carbon catabolite repression, Hfq and Crc link key metabolic and virulence processes in *Pseudomonas* species. The two proteins affect biofilm formation, motility (*O'Toole et al., 2000*, *Huang et al., 2012*; *Zhang et al., 2012*; *Pusic et al., 2016*), biosynthesis of the virulence factor pyocyanin (*Sonnleitner et al., 2003*; *Huang et al., 2012*), and antibiotic susceptibility (*Linares et al., 2010*; *Heitzinger, 2016*). Recent ChiP-seq studies indicate that Hfq and Crc have an even broader regulatory impact in *Pseudomonas* and can work in concert to bind many nascent transcripts co-translationally, uncovering a large number of potential regulatory targets (*Kambara et al., 2018*). Crc is a specialized protein that is found in a limited subset of bacteria including Pseudomonads but excluding Enterobacteriaceae such as *E. coli*. The Crc protein is a member of the EEP family (Exonuclease/Endonuclease/Phosphatase), but the absence of key catalytic residues in the active site cleft indicate that Crc has lost enzymatic function (*Milojevic et al., 2013*).

Crc and Hfq act together to translationally repress mRNA target genes. The targets are then subjected to degradation, which might trigger disassembly of the Hfq/Crc/RNA complex (*Sonnleitner and Bläsi, 2014*). The impact of Hfq/Crc on translational repression is countered by the non-coding RNA CrcZ, which increases in levels when preferred carbon sources are exhausted (*Sonnleitner et al., 2009*; *Sonnleitner and Bläsi, 2014*). CrcZ expression is under control of the alternative sigma factor RpoN and the two-component system CbrA/B (*Sonnleitner et al., 2009*). Although the signal responsible for CbrA/B activation remains unknown, it is thought to be related to the cellular energy status (*Valentini et al., 2014*).

The mechanism by which Hfq and Crc act in translational repression of target mRNAs involves binding of both proteins to ribosome-binding sequences. In the case of the translationally repressed *amiE* mRNA, encoding aliphatic amidase, the distal face of Hfq binds to an A-rich segment termed $amiE_{6ARN}$ that comprises the ribosome binding site (*Sonnleitner and Bläsi, 2014*). In contrast, Crc has no intrinsic RNA-binding activity (*Milojevic et al., 2013*) but strengthens binding of A-rich target transcripts to the distal side of Hfq (*Sonnleitner et al., 2018*). How Hfq binds to A-rich RNAs is structurally understood (*Link et al., 2009*), but the cooperative role of Crc in this context has not been elucidated. To gain insight into how *P. aeruginosa* Hfq cooperates with Crc in translational repression of *amiE*, we determined the structure of the complex they form on the Hfq binding motif of the CCR-controlled *amiE* mRNA using cryo-electron microscopy (cryoEM). Our analyses revealed that the components form higher order assemblies and explain for the first time how a widely occurring structural motif can support the association of Hfq and RNA into cooperative ribonucleoprotein complexes that have regulatory roles. We observe that the interactions supporting the quaternary structure are required for in vivo translational regulation. These findings expand the paradigm for in vivo action of Hfq through cooperation with the Crc helper protein and RNA to form effector assemblies.

## Results

### An ensemble of Hfq/Crc/$amiE_{6ARN}$RNA assemblies

For cryo-EM structural studies of the Hfq/Crc/RNA complex, purified recombinant Hfq and Crc proteins were mixed with an 18 nucleotide Hfq binding motif from the translation initiation region of the CCR-controlled *amiE* mRNA (hereafter termed $amiE_{6ARN}$). The defined sequence (5'-AAA-AAU-AAC-AAC-AAG-AAG-3') has a binding motif comprised of 6 repeats of an A-R-N pattern preferred by the distal face of Hfq (*Sonnleitner and Bläsi, 2014*). The 18 nucleotides of $amiE_{6ARN}$ could be distinguished in the cryoEM maps and will be referred to with numbers 1–18 from the 5′ to the 3′ end. The purified sample of Hfq/Crc/$amiE_{6ARN}$, after mild chemical crosslinking, yielded well defined single particles on graphene oxide in thin, vitreous ice. Analysis of the reference free 2D class averages and subsequent 3D classification indicated three principal types of complexes corresponding to different stoichiometries of Hfq (hexamer):Crc:$amiE_{6ARN}$ with compositions 2:2:2, 2:3:2 and 2:4:2 (*Figure 1*). These higher order assemblies are in agreement with recently observed SEC-MALS and mass spectrometry results which excluded a simple 1:1:1 assembly (*Sonnleitner et al., 2018*). The maps for the complexes are estimated to be 3.1 Å, 3.4 Å and 3.2 Å in resolution, respectively, based on gold-standard Fourier shell correlations (*Figure 1—figure supplement 1*). The distribution of the complexes corresponds to roughly 49%, 29% and 23% for the 2:2:2, 2:3:2 and 2:4:2 complexes (*Figure 1*). The individual crystal structures of Hfq and Crc dock well into the cryoEM densities (*Figure 1* and *Figure 1—figure supplement 2*), and aside from side chain rotations there are few other structural changes of the components (*Figure 1—figure supplement 2*). To probe for artefacts introduced by the crosslinking experiments, CryoEM analyses were subsequently performed with samples that had not been treated by crosslinking using grid preparation conditions optimised for the crosslinked specimens. Although limited to lower resolution, data from these specimens revealed that the quaternary structure remained unchanged within the resolution limits of the data for the model being compared (*Supplementary file 1A*).

### A specialised and recurring RNA conformation in Hfq-mediated regulation

*Link et al. (2009)* described the crystal structure of *E. coli* Hfq bound to a polyriboadenylate 18-mer and observed that the RNA encircled the distal face of the Hfq hexamer *via* a repetitive tripartite binding scheme (*Figure 2—figure supplement 1A*). Each base triplet is partially embedded between adjacent Hfq monomers and is mostly surface exposed, folding into a' crown-like' conformation. We observe striking similarities with the fold of the authentic $amiE_{6ARN}$ species on the distal side of the *P. aeruginosa* Hfq hexamer (*Figure 2* and *Figure 2—figure supplement 1A*). Notably, the cryoEM maps were calculated without any reference to *Link et al. (2009)*. The agreement between the co-crystal structure of the analogous complex and the entirely independently derived cryoEM based model provides thus a strong validation.

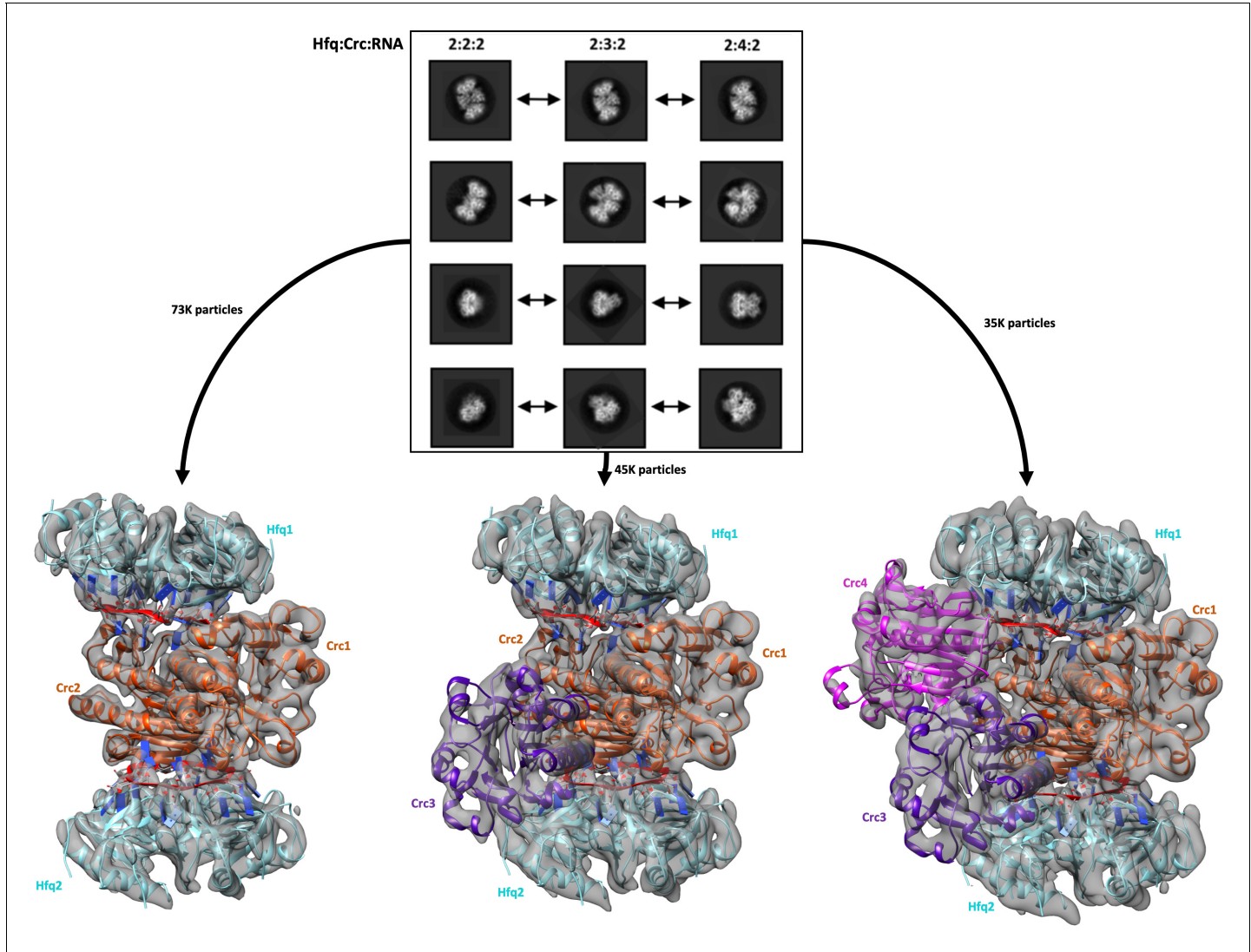

**Figure 1.** Reference free 2D classification and 3D classification of Hfq/Crc/RNA particles. Three main classes of particles were observed after reference-free 2D classification (top), corresponding to Hfq/Crc/$amiE_{6ARN}$ assemblies with compositional stoichiometries of 2:2:2, 2:3:2 and 2:4:2. The $amiE_{6ARN}$ species (red) constitute the main interaction interface between Hfq and Crc, together forming the 2:2:2 core complex observable in all three models (bottom). Cyan: Hfq hexamer, orange purple and pink: Crc monomers, red: $amiE_{6ARN}$. All cryoEM maps were low-pass filtered to 6 Å for clarity of presentation and the crystal structures were docked in as rigid bodies. Only a subset of the high quality 2D classes are shown in the panel.

DOI: https://doi.org/10.7554/eLife.43158.003

The following figure supplements are available for figure 1:

**Figure supplement 1.** Resolution of maps and angular distributions of 2D images.
DOI: https://doi.org/10.7554/eLife.43158.004

**Figure supplement 2.** Raw micrograph image, fit of refined model into cryo-EM map, and correlation of experimental map and models.
DOI: https://doi.org/10.7554/eLife.43158.005

Like its *E. coli* homologue, *Pseudomonas* Hfq contains six tripartite binding pockets on the distal side, capable of binding a total of 18 nucleotides. Each of the six RNA triplets of the $amiE_{6ARN}$ RNA fits into an inter-subunit cleft in Hfq (*Figure 2*). The specific, star-shaped RNA fold is guided by six positively charged protuberances on the distal face of Hfq, with the phosphate backbone circularly weaving in between these, seemingly to minimise steric hindrance while maximizing surface interactions (*Figure 2*). As described by *Link et al. (2009)*, each pocket consists of an adenosine specificity site (A), a purine nucleotide specificity site (R), and a presumed RNA entrance/exit site (N) which is non-discriminatory (*Figure 2—figure supplement 1A*). Hfq thus has a structural preference for

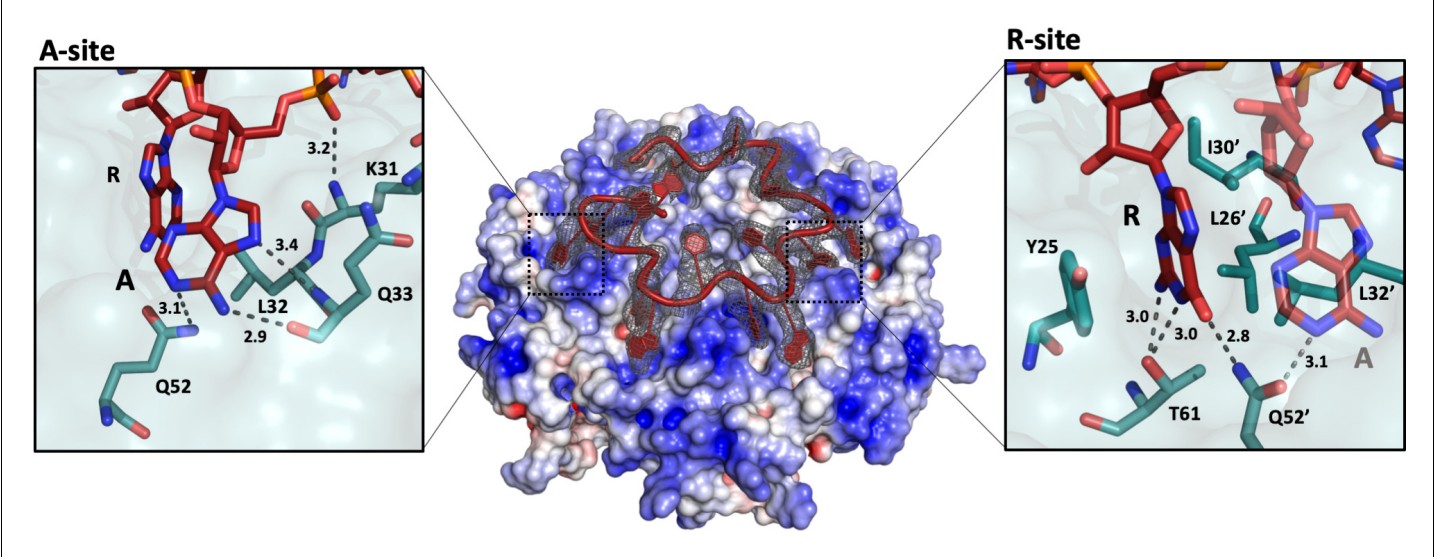

**Figure 2.** The 'A-R-N crown' in the Hfq/*amiE*$_{6ARN}$ RNA complex. 6 RNA triplets are partially embedded in six binding pockets on the Hfq distal side, forming a weaving, crown-like pattern. The A and R sites are occupied by adenine and a purine, respectively, whereas the RNA entry/exit site has no discriminatory preferences and is therefore referred to as the 'N' site. The Cryo-EM density for *amiE*$_{6ARN}$ is depicted as a grey mesh, with the RNA 'crown' modelled in red. Positively charged protuberances (blue) guide the RNA to fold into a star-shaped conformation to maximize the surface interaction between the negatively charged RNA backbone, and the positively charged Hfq surface pattern. An atomic model of the A-R-N occupation pattern. Left panel: Adenosine specificity site. Right panel: Purine specificity site. *amiE* nucleotide carbon atoms are depicted in red, Hfq carbon atoms are in green.

DOI: https://doi.org/10.7554/eLife.43158.006

The following figure supplement is available for figure 2:

**Figure supplement 1.** Interactions of RNA with the distal face of Hfq.

DOI: https://doi.org/10.7554/eLife.43158.007

(ARN)$_n$ RNA stretches on its distal side, where N is any nucleotide. The adenosine specificity (A) sites are organised identically to the corresponding A sites in *E. coli* Hfq, forming hydrogen bonds between the peptide backbone and carboxyl-groups of Gln33 and the N6,7 atoms of the adenosine base, and a polar interaction between Gln52 (N$_\epsilon$) and the N1 atom of the adenosine base (*Figure 2*, *Figure 2—figure supplement 1A*). The peptide backbone amide of residue Lys31 interacts with the 5' phosphate group of adenine. Finally, the adenine base is stacked against the side chain of Leu32 (*Figure 2*). The interactions with N1, by the carbonyl of Q33, and N7, by the amide of Q33, confer pocket-specificity for A as they are not compatible with a G, which would form a repulsive contact *via* its O6 and peptide carbonyl (*Figure 2*). The purine (R) specificity site is defined by two neighbouring monomers, where the side chains from Tyr25 and from Leu26', Ile30' and Leu32' (where the prime denotes residues from a neighbouring subunit) contact the nucleotide aromatic base. In *amiE*$_{6ARN}$, one R-site is populated by a guanine, forming a hydrogen bond between the N$_\epsilon$ of Gln52' and the guanine exocyclic O6 (*Figure 2*). Just like in the *E. coli* Hfq/polyA$_{18}$ structure (*Link et al., 2009*), Gln52' forms a physical link between the A and R sites. Previous structures were obtained from polyA RNA, whereas the structures presented here were solved with the authentic *amiE* Hfq recognition site. Interestingly, Thr61 O$_\gamma$ forms a double hydrogen bond with the N1 and the exocyclic N2 from the guanine base, which was not seen previously (*Link et al., 2009*) as all R-sites were occupied by adenine residues (*Figure 2*). As we will describe further below, the N site base interacts with Crc, but without apparent sequence preference. Strikingly, the A-R-N motif is not supported by Hfq of the Gram-positive bacteria *Staphylococcus aureus* and *Bacillus subtilis*, which instead use a R-L (purine, linker) motif (*Horstmann et al., 2012*) (*Figure 2—figure supplement 1B*). Phe 30 hinders the RNA backbone, and thus the A-site base, from entering the A-site, while stabilising the R-site together with M32 *via* stacking interactions.

## The quaternary structure of the core complex

In the core complex (with 2:2:2 stoichiometric composition of Hfq:Crc:RNA), two Hfq hexamers sandwich the RNA and Crc (*Figure 3A*), with each Hfq interacting with one $amiE_{6ARN}$ RNA and two Crc components. The assembly has C2 symmetry, with the molecular twofold axis passing through the interface of the two Crc molecules. The Crc´s in the assembly self-interact in the same way as observed in the crystal structure of the isolated Crc dimer, where the interface is generated through crystallographic symmetry (*Wei et al., 2013*; *Milojevic et al., 2013*). As anticipated from mutational analyses (*Sonnleitner et al., 2018*), the dominating protein/RNA interaction is made by the distal face of Hfq, forming an interface area of roughly 2270 $Å^2$.

The two Crc molecules interact with RNA residues exposed on the surface of Hfq, and both Crc molecules contact the Hfq-rim on the distal side (*Figure 3A*). The Crc protomers form contacts mainly with the backbone phosphate groups and exposed ribose rings of the RNA (*Figure 3A and C*). Because the Crc forms antiparallel dimers, there are two modes of interaction of the Crc with $amiE_{6ARN}$ RNA. In one mode of interaction, Arg140 $\eta^1$-NH$_2$ and Arg141 $\varepsilon$-NH and $\eta^1$-NH$_2$ interact with the phosphodiester backbone of $amiE_{6ARN}$. Arg140 and Arg196 form a sandwich with the purine-base of the A3 nucleotide at an Entry/Exit 'N' site of $amiE_{6ARN}$ RNA. Interestingly, Arg140 participates in the Crc1-Crc2 dimer interface in the Crc-dimer as solved by crystallography *via* contacts with E193 (pdb-ID 4jg3, data not shown), but our maps clearly show a rotamer shift of the R140 side chain away from E193, forming contacts with *amiE* as described above. The Arg196 $\varepsilon$-NH and $\eta^1$-NH$_2$ groups form hydrogen bonds with the U6-$amiE_{6ARN}$ backbone and the U6 O2 group forms a hydrogen bond with the Met156 amide. In the second mode of interaction, Lys155 $\zeta$-NH$_2$ makes a hydrogen bond with the OP$_2$-group of C9 and the ribose hydroxyl group. Additional hydrogen bonds are formed between Trp161 $\varepsilon^1$-NH and Arg162 $\eta^{1/2}$-NH$_2$ and the phosphate backbone of $amiE_{6ARN}$ (*Figure 3A*).

The highly organised interactions in the core complex (*Figure 3A*) illustrate how the bases of $amiE_{6ARN}$ as presented by Hfq constitute a molecular interface for the RNA-mediated interactions between Hfq and Crc. Crc forms small contact surfaces to the RNA, to Hfq, and to itself as a homodimer; these small areas work together to give an assembly that is most likely stabilised through chelate cooperativity. Notably, there is a striking absence of any lower order assemblies in the cryo EM micrographs. The 2:2:2 complex is therefore likely to be the minimal complex formed when all components are present and must be constructed in an 'all or nothing' manner, somewhat like a binary switch.

## Function, origins and validation of subunit cooperativity in the 2:2:2 complex

The dimer interface of the Crc pair is the largest protein-protein interface in the 2:2:2-complex and has a buried area of 766 $Å^2$, which typically corresponds to a moderate intermolecular affinity. The key dimerization interface is maintained by salt bridges between Arg229-Arg230 of one Crc monomer and Glu142 of the second Crc monomer, which is further stabilised by pi-stacking of the Phe231-Phe231 rings across the twofold axis (*Figure 3A*). The phenylalanine residues are in turn stabilised by stacking interactions with Trp255 of the same Crc monomer (not shown). Two additional polar contacts are formed between Arg137 and the Asn184 carbonyl group of two pairs of helices in the Crc dimer, forming a smaller secondary interface (*Figure 3A*).

To verify selected interactions between the Crc protomers and Crc and RNA, we explored the effects of Crc variants on Hfq/Crc-mediated repression of an *amiE::lacZ* translational reporter gene encoded by plasmid pME9655 (*Sonnleitner et al., 2018*). In addition, an in vitro co-immunoprecipitation assay was employed to assess the capacity of Hfq and Crc variants to form a complex in the presence of $amiE_{6ARN}$ RNA (*Sonnleitner et al., 2018*). First, we asked whether R140 (Crc$_{R140}$) is required for the interaction of the protein with the RNA (*Figure 3A*, bottom left inset). As shown in *Figure 3B*, the Crc$_{R140E}$ mutant was deficient in translational repression of the *amiE::lacZ* reporter gene, similarly as observed for the *crc* deletion strain. Moreover, Crc$_{R140E}$ did not co-immunoprecipitate with Hfq in the presence of $amiE_{6ARN}$ RNA (*Figure 3—figure supplement 1A*), indicating that the interaction between Crc$_{R140}$ and RNA is pivotal for Hfq/Crc/RNA complex formation.

Next, we focused on the interactions in the E142 and R229/R230 'triangle' (*Figure 3A*, top left inset) for the Crc-Crc interaction. The single mutant proteins Crc$_{R229E}$ and Crc$_{R230E}$ did not affect

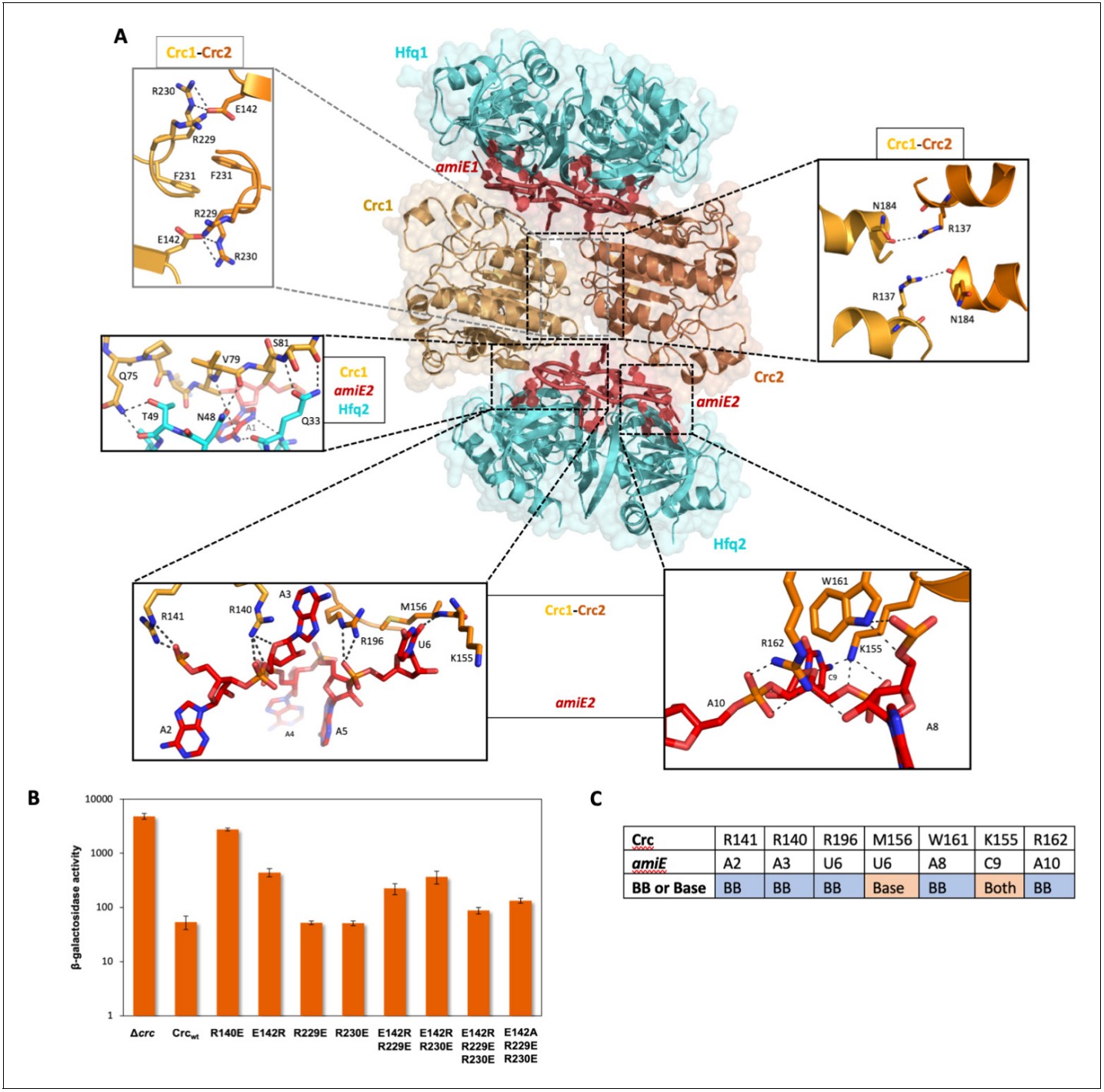

**Figure 3.** Model of the 2:2:2 Hfq/Crc/RNA complex and validation of interactions. (**A**) Atomic model of the 2:2:2 Hfq/Crc/RNA complex. The view is along the C2 molecular symmetry axis which passes through the homodimeric Crc interface. Hfq hexamers flank the Crc dimer and present the $amiE_{6ARN}$ RNA to form two different interfaces with the Crc protomers, which form an anti-parallel dimer. The Crc protomers form strong polar contacts with mainly the backbone phosphate groups and exposed ribose rings (bottom insets). Two small C2 symmetric binding interfaces constitute the Crc dimerisation (top insets). A single short stretch on each Crc monomer binds a Hfq monomer (middle left panel). Dimeric Crc in yellow and orange, $amiE_{6ARN}$ RNA in red, Hfq hexamers in cyan. (**B**) Translational regulation of an *amiE::lacZ* reporter gene by Crc variants. *P. aeruginosa* strain PAO1Δ*crc* (pME9655) harboring plasmids pME4510 (vector control), pME4510crc_Flag (Crc_wt) or derivatives thereof encoding the respective mutant proteins was grown to an OD_600 of 2.0 in BSM medium supplemented with 40 mM succinate and 40 mM acetamide. The β-galactosidase values conferred by the translational *amiE::lacZ* fusion encoded by plasmid pME9655 in the respective strains are indicated. The results represent data from two independent experiments and are shown as mean and range. (**C**) Table of Crc-*amiE* contacts in the 2:2:2 assembly. BB: Crc contacts with the RNA backbone; Base: Crc contacts with RNA bases.

*Figure 3 continued on next page*

*Figure 3 continued*

DOI: https://doi.org/10.7554/eLife.43158.008

The following figure supplement is available for figure 3:

**Figure supplement 1.** Association of Hfq and Crc in the presence of RNA.

DOI: https://doi.org/10.7554/eLife.43158.009

translational repression of *amiE::lacZ*, whereas the function of the Crc$_{E142R}$ variant was diminished (*Figure 3B*), indicating that E142 can interact with either R229 or R230. The de-repression of *amiE: lacZ* observed with the Crc$_{E142R}$ variant was partially compensated by the double mutant proteins Crc$_{E142R, R229E}$ and Crc$_{E142R, R230E}$ (*Figure 3B*). In addition, the Crc$_{E142R}$ and Crc$_{R230E}$ variants were impaired in Hfq/Crc/RNA complex formation as shown by the co-immunoprecipitation assay (*Figure 3—figure supplement 1A*). Strikingly, the compensatory changes present in the triple mutant protein Crc$_{E142R, R229E, R230E}$ almost fully restored translational repression of the *amiE::lacZ* reporter gene (*Figure 3B*). As the respective Crc variant proteins were produced at comparable levels (*Figure 3—figure supplement 1B*), these mutational studies support the in vivo role for the interactions of the Crc protomers observed in the cryo-EM models. The exchange of Glu142 with an Ala in the triple mutant protein Crc$_{E142A, R229E, R230E}$ restored as well translational repression of the *amiE::lacZ* reporter gene (*Figure 3B*). We interpret this as showing that the rescue of the deleterious mutations does not require charge compensation, but removing any potential charge repulsion.

## Recruitment of additional Crc units to the core

The quaternary organisation of the 2:2:2 complex forms a core unit that is also present in the 2:3:2 and 2:4:2 complexes. In that common core, the interaction of Crc with RNA leaves approximately half of the accessible surface of the nucleic acid exposed. For the 2:3:2 and 2:4:2 complexes, additional Crc units are recruited through interactions with the exposed portion of the RNA. As such, the C2 symmetry is broken by the third Crc molecule in the 2:3:2 complex (*Figure 1*). Interestingly, recruitment of a fourth Crc monomer to the complex restores the C2 symmetry, preserving the symmetry axis from the core complex, but with a conformationally different Crc dimer interface between Crc molecules 3 and 4 (*Figure 4A*). The two additional Crc monomers have small surface-area contacts with the rest of the complex and are likely to be comparatively mobile, which may account for the stronger variation in resolution for the 2:3:2 and 2:4:2 maps compared to the rather rigid 2:2:2 core assembly (*Figure 1—figure supplement 1*).

## Function, origins and validation of subunit cooperativity in the 2:4:2 complex

The protomer interactions of the 2:2:2 assembly are highly interdependent, and once the core complex is generated it can apparently recruit additional Crc molecules, forming the 2:3:2 and 2:4:2 complexes. In the 2:4:2 complex, a second type of Crc dimer seems to assemble with a smaller buried surface (*Figure 4A*). Such additional dimers can only form when an intact 2:2:2 core complex is present, as they are not observed in the core complex itself. Notably, the additional Crc dimer is in a more 'open' conformation than the dimer in the core (*Figure 4C*). The same key Crc dimer interface is occupied but seems to serve as a dynamic hinge, whereas the secondary, smaller, dimer interface between the Crc helices is absent to allow the new Crc dimer to adopt an 'open' conformation. Arg230 is reorganised by Glu193 in the same protomer to self-interact with the corresponding Arg230 in the partner Crc, rather than with Glu142 (*Video 1*). Additional hydrogen bonds are formed between Arg233 and Glu193, whereas Arg229 is no longer part of the dimer interface (*Figure 4A*). Both Arg230 and Glu193 seem to play important roles in providing the structural freedom to form a dynamic hinge (*Figure 4C*). Glu193 is part of the Crc1-Crc2 interface in the crystal structure but does not participate in this interface when assembled in the 2:2:2 core complex. Instead it aids in coordinating the Crc3-Crc4 interface in the 2:4:2 assembly. Interestingly, Crc3 and Crc4 do not interact with Crc1 or Crc2, and seem to be recruited solely *via* Hfq/*amiE*$_{6ARN}$. By reorganising the extra Crc molecules 3 and 4 that bind the 2:2:2 core (*Figure 4A*), the alternative Crc dimer is able to utilise one of two basic patches on its surface when engaging *amiE*$_{6ARN}$ without causing steric hindrance to the already bound Crc dimer.

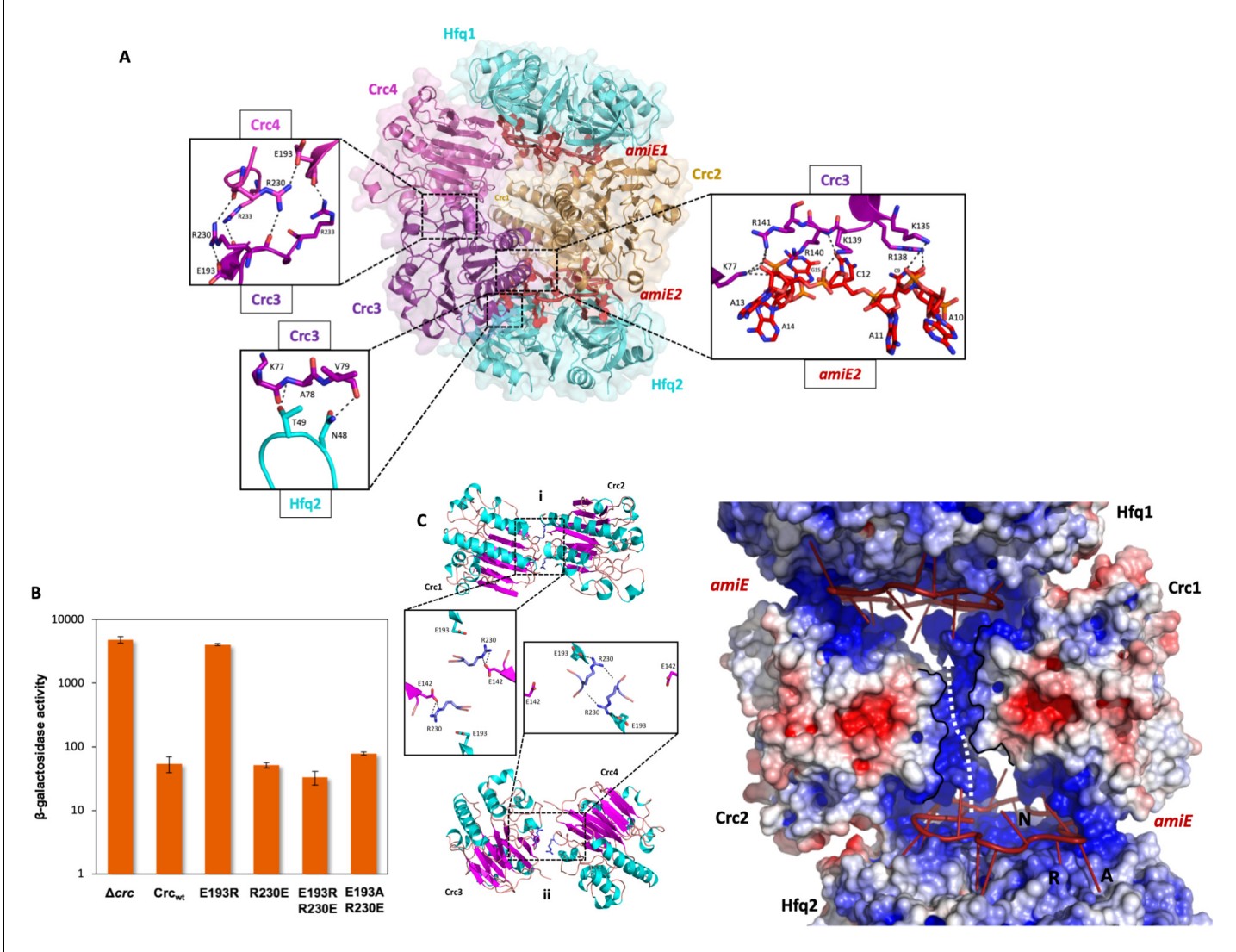

**Figure 4.** Model of the 2:4:2 Hfq/Crc/RNA complex and validation of interactions. (**A**) Atomic model of the 2:4:2 Hfq/Crc/RNA complex. The insets show additional Hfq-Crc, Crc-Crc and Crc-RNA interactions not present in the 2:2:2 complex. The Crc3-4 dimer is formed by only one interface, with an R230-R230 interaction at the core, which globally overlaps with the dimer interface of the Crc1-Crc2 dimer (top left inset). Only one of two RNA-binding patches is presented to $amiE_{6ARN}$ in the Crc3-4 dimer, yet exploited more extensively (right inset). A small interface is formed between Crc3-4 and Hfq. Crc dimer in yellow, $amiE_{6ARN}$ RNA in red, Hfq hexamers in cyan, extra Crc dimer in magenta and purple. (**B**) Translational regulation of an $amiE::lacZ$ reporter gene by Crc variants, as described in *Figure 3B*. The results represent data from two independent experiments and are shown as mean and range. (**C**) Two distinct dimeric Crc species are observed in the three complexes solved by cryoEM. i: The self-complementary interaction of the 2:2:2 core complex. ii: In the 2:4:2 complex, an alternative dimer is formed, showing a twisted dimer interface and more open configuration, with Arg230 serving as a dynamic hinge (bottom). (**D**) An electropositive half-channel runs along the dimer interface of the Crc1-2 dimer, and in the context of the Hfq/Crc/RNA assembly it could potentially serve as a conduit for RNA (dotted white arrow; see *Figure 5*). The A, R, and E RNA interaction sites of Hfq are annotated.

DOI: https://doi.org/10.7554/eLife.43158.010

The Arg233-Glu193 interaction is unique for the 2:4:2 assembly and was therefore assessed in vivo. Strikingly, the $Crc_{E193R}$ mutation fully abrogated translational repression of the $amiE:lacZ$ reporter gene (*Figure 4B*). The model predicts that the deleterious $Crc_{E193R}$ mutation can be compensated by the substitution of $Crc_{R230E}$ to re-establish the interaction. This pair does indeed behave as predicted, as the $Crc_{E193R, R230E}$ variant restored translational repression of the $amiE::lacZ$ reporter gene, further confirming the in vivo importance of the 2:4:2 assembly during CCR (*Figure 4B*). The exchange of Glu193 with Ala in the $Crc_{E193A, R230E}$ mutant protein also restored

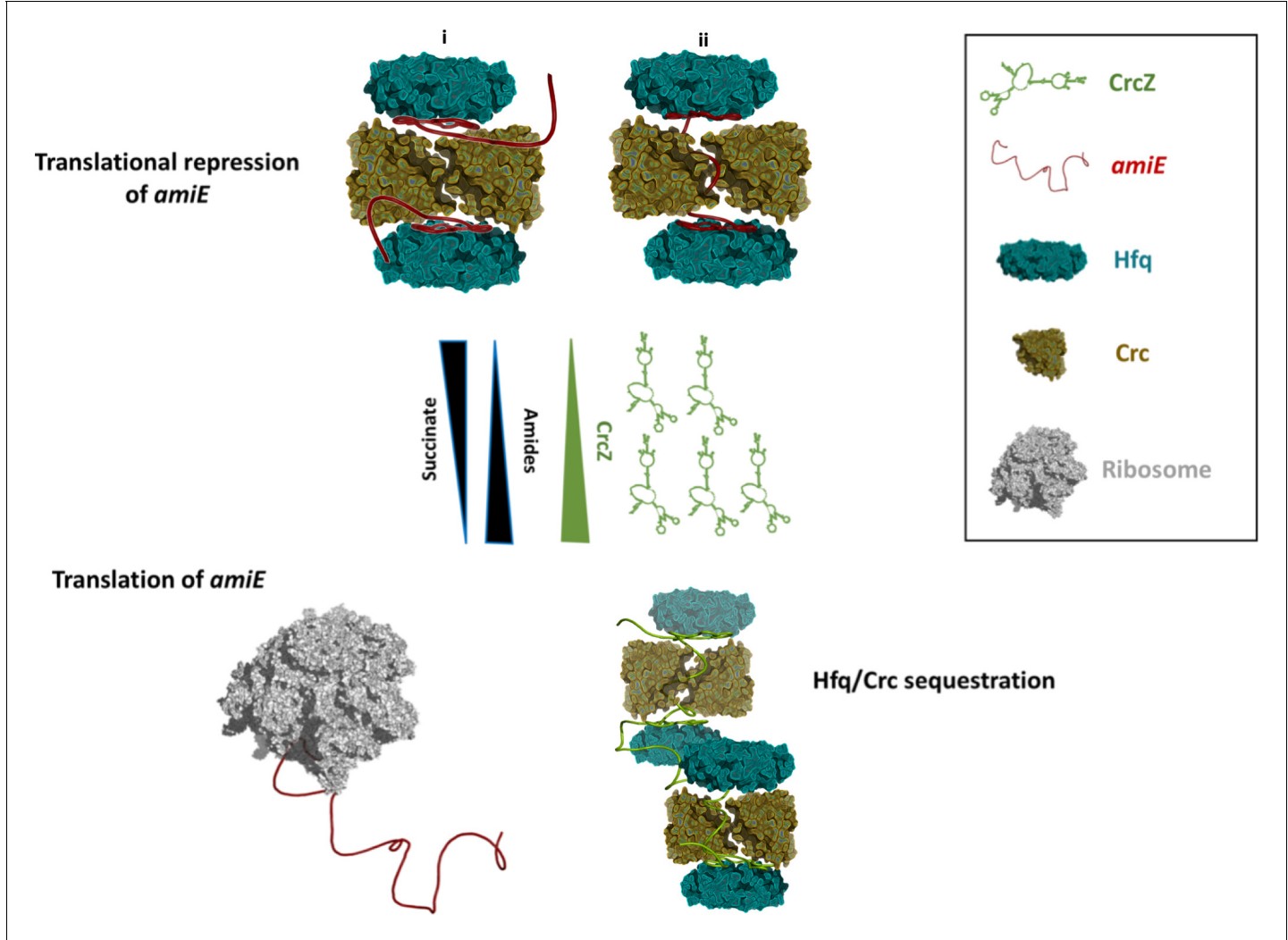

**Figure 5.** Schematic pathway of carbon catabolite repression. When the preferred carbon source, succinate, is abundant, cellular CrcZ levels are low and Hfq and Crc occlude the *amiE* ribosome binding site by forming a higher order assembly, rendering CCR active and repressing synthesis of for example aliphatic amidase (top). We envision that translational repression may either occur on two mRNA molecules (i) or on a single mRNA molecule that comprises another Hfq binding site downstream of the first (ii). Upon depletion of succinate, CrcZ levels increase and CrcZ sequesters Hfq and Crc from *amiE*, potentially by occupying the multiple A-R-N patches on CrcZ and forming multicomponent 'beads on a string' (bottom). As such CCR is abolished, allowing metabolism of a secondary carbon source, for example amide conversion by aliphatic amidase.
DOI: https://doi.org/10.7554/eLife.43158.012

translational repression of the reporter gene, albeit to a somewhat reduced extent when compared with the Crc$_{E193R, R230E}$ variant (*Figure 4B*). Again, these results suggest that the rescue of the deleterious mutations does not require charge compensation, but act by removing any potential charge repulsion.

In addition to Crc Arg140 and Arg141, Crc K139 ζ-NH$_2$ makes a hydrogen bond with the OP$_2$-group of A12, Arg138 η$^1$-NH$_2$ interacts with the ribose hydroxyl group of C9 and K135 ζ-NH$_2$ forms a hydrogen bond with the A11 OP$_2$. Finally, the O2 of cytosine C12 engages in a hydrogen bond with the backbone amino group of Arg140. Direct interactions between the reorganised Crc dimer and Hfq are limited to the same Crc β-strand and exposed loop of a sole Hfq monomer, as in the core complex. Due to the open conformation of the alternative Crc dimer, the Hfq Thr49 hydroxyl group now forms a hydrogen bond with the Ala 78 amide group (*Figure 4A*).

Interestingly, a basic half-channel is formed over the core dimer interface, with additional basic patches spread over the RNA binding surface of the Crc dimer (*Figure 4D*). Speculatively, longer

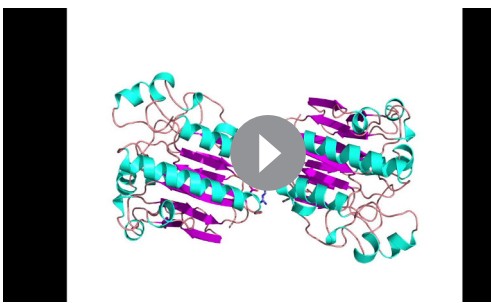

**Video 1.** A comparison of the Crc-Crc interface in the 2:2:2 Hfq:Crc:RNA complex (the core complex) and the second type of interface formed between the additional Crc components of the 2:4:2 complex.
DOI: https://doi.org/10.7554/eLife.43158.011

RNA species could travel though the surface exposed half-channel and interconnect all components of the core complex into a highly organised assembly on this target RNA (*Figure 5*).

## Discussion

Many functional studies have highlighted the cooperation of global posttranscriptional regulators in controlling the fate of targeted transcripts. A major gap in our current understanding has been the lack of high-resolution structural data of these highly coordinated cellular processes. Here, we report for the first time structural information on how a partner protein can assist the role of Hfq in direct translational repression when bound to a translational initiation region ($amiE_{6ARN}$) by forming a multi-component assembly. The added value of the Crc in the complex as compared to recognition of the A-rich sequence by Hfq is the potential for highly cooperative switches for activation/repression and greater specificity.

The *amiE* RNA translation control site is an A-rich fragment that occupies almost entirely the distal surface of the *P. aeruginosa* Hfq, weaving in between basic, surface exposed islands (*Figure 2*). There are striking similarities to the structure of the $polyA_{18}$ complex with *E. coli* Hfq (*Link et al., 2009*), which greatly added to the understanding of RNA binding and chaperone mechanisms, and hinted at how the distinct polyA RNA interaction might enable Hfq-mediated regulation. The $polyA_{18}$/Hfq structure revealed rules for recognition of motifs of the type A-R-N, where R is purine and N is any base (*Figure 2—figure supplement 1*). The *P. aeruginosa* Hfq interaction with $amiE_{6ARN}$ follows the same rules. The A-R-N repeat occurs in many RNAs, and it is a recurring motif in the nascent transcripts that associate with Hfq and Crc in *Pseudomonas* (*Kambara et al., 2018*). It has been proposed that the exposed bases (at the Entry/Exit site, corresponding to 'N' in the A-R-N motif) could mediate RNA to RNA interactions (*Schulz et al., 2017*), but we observe that the exposed bases are presented for protein recognition. The exposed bases at the 'N' position and RNA backbone in the Hfq/$amiE_{6ARN}$ complex are available for interactions with Crc to form a cooperative assembly (*Figures 3* and *4*) that mediates translational repression on target mRNAs, that is carbon catabolite repression in vivo when the preferred carbon source is available. Strikingly, the A-R-N motif is not supported by the Hfq of the Gram-positive bacteria *Staphylococcus aureus* and *Bacillus subtilis* (*Horstmann et al., 2012*) (*Figure 2—figure supplement 1B*). In view of this divergence, it seems unlikely that Hfq in the Gram-positive species form assemblies that resemble the *P. aeruginosa* Hfq/Crc assembly.

Previous studies have shown that both Hfq and Crc are required for tight translational repression of mRNAs, which are subject to carbon catabolite repression (CCR) (*Sonnleitner and Bläsi, 2014*; *Moreno et al., 2015*). The presence of Crc did not significantly enhance the affinity of Hfq for $amiE_{6ARN}$ RNA (*Sonnleitner et al., 2018*). However, the simultaneous interactions of Crc with both binding partners resulted in an Hfq/Crc/RNA assembly with increased stability when compared with the Hfq/RNA complex alone (*Sonnleitner et al., 2018*). In light of our structural studies, the enhancing effect of Crc in Hfq-mediated translational repression of target mRNAs during CCR (*Sonnleitner and Bläsi, 2014*; *Moreno et al., 2015*) can be explained by sandwiching the Hfq binding site on mRNA between both binding partners. Thus, the structural model can rationalize the observed increased stability of the Hfq/Crc/$amiE_{6ARN}$ assembly when compared to the sole Hfq/$amiE_{6ARN}$ complex (*Sonnleitner et al., 2018*). It is conceivable that full repression is only achieved when $amiE_{6ARN}$ is masked entirely in the 2:4:2 complex, which is supported by our in vivo studies.

The question arises why a higher order assembly such as the 2:2:2 core is formed and not a simpler complex. The structural data indicate that the dimerization of Crc provides the key step for formation of the 2:2:2 complex, because it will pre-organise a copy of the surface that interacts with a binary Hfq/RNA complex so that a second Hfq/RNA complex can be recruited. Thus, all components seem to be necessary to form the complex so that there is no formation of lower order 'sub

assemblies'. The structural data are consistent with Crc having no capacity for RNA binding by itself (*Milojevic et al., 2013*). The Hfq/Crc/RNA complex may thus be assembled in a checklist-like manner through numerous small contacting surfaces and when the RNA target is presented by Hfq in a specific, well-defined configuration. In this way, the components interact mutually through chelate cooperative effects. It seem likely that the 2:2:2 core is formed first followed by recruitment of the other Crc components.

We envisage that the 2:2:2 core and higher order assemblies might interact with other mRNAs. The higher order assembly could capture two of such mRNA substrates as shown in *Figure 5i*, but chelate effects might instead induce formation of the complex on a single mRNA target. In that scenario, a portion of the mRNA could thread through the central basic half channel (*Figure 4D*) as depicted in *Figure 5ii* and potentially recruit the second Hfq hexamer. In this context it is worth noting that several mRNAs, including *amiE*, which are directly repressed by Hfq/Crc comprise another putative Hfq binding motif downstream of the start codon (*Sonnleitner et al., 2018*). Therefore, we are currently exploring the possibility whether this second Hfq binding motif contributes to Hfq/Crc assembly on longer mRNA substrates and whether it likewise may confer specificity for these Hfq/Crc repressed substrates.

Under conditions of catabolite repression regulation, pull-down assays showed that Hfq and Crc form a co-complex in the presence of the 426nt long CrcZ RNA (*Moreno et al., 2015*; *Sonnleitner et al., 2018*). In the presence of less preferred carbon sources, the expression levels of CrcZ RNA increase (*Sonnleitner et al., 2009*) and CrcZ functions as an antagonist in Hfq/Crc mediated translational repression of catabolic genes. The CrcZ RNA has multiple A-R-N triplets that could be sites for Hfq/Crc interaction (*Sonnleitner and Bläsi, 2014*) and sequester multiple Hfq/Crc proteins (*Figure 5*). Thus, under conditions where CCR is relieved, CrcZ RNA would serve as a sponge for Hfq/Crc to prevent repression of genes encoding proteins required for the utilization of less preferred carbon sources (*Figure 5*). How the CrcZ RNA is displaced from Hfq/Crc remains unknown. However, the assemblies are likely to be dynamic and the displacement process might resemble that proposed for the step-wise exchange of sRNAs on Hfq (*Fender et al., 2010*).

Recent findings show that the regulatory spectrum of Hfq and Crc is much broader than initially expected. Hfq was found to bind more than 600 nascent transcripts co-transcriptionally often in concert with Crc (*Kambara et al., 2018*). These findings indicate that Hfq and Crc together regulate gene expression post-transcriptionally beyond just catabolite repression. Understanding how gene expression is regulated post-transcriptionally in pathogens such as *P. aeruginosa* may provide potential targets for novel drug design. Hfq and Crc are involved in key metabolic and virulence processes in *Pseudomonas* species (*O'Toole et al., 2000*; *Sonnleitner et al., 2003*; *Sonnleitner et al., 2006*; *Linares et al., 2010*; *Huang et al., 2012*; *Zhang et al., 2012*; *Zhang et al., 2013*; *Sonnleitner and Bläsi, 2014*; *Pusic et al., 2016*). Disrupting the interface of the core assembly of the Hfq/Crc complex might be one strategy to counter, among other, metabolic regulation and consequently its downstream processes that impact on virulence during infection. A recent study showed how overproduction of the aliphatic amidase AmiE strongly reduced biofilm formation and almost fully attenuated virulence in, amongst others, a mouse model of acute lung infection (*Clamens et al., 2017*). Novel drugs that specifically counteract Hfq/Crc/*amiE* assembly formation and prevent repression of AmiE production could induce the phenotype described by *Clamens et al. (2017)*. The high resolution structures presented here provide a starting point for novel strategies to interfere with carbon regulation in a pathogenic bacterium for therapeutic intervention of threatening infections.

## Materials and methods

**Key resources table**

| Reagent type (species) or resource | Designation | Source or reference | Identifiers | Additional information |
|---|---|---|---|---|
| Gene (*Pseudomonas aeruginosa*) | *crc* | NA | Pseudomonas genome database: PA5332 | |

*Continued on next page*

*Continued*

| Reagent type (species) or resource | Designation | Source or reference | Identifiers | Additional information |
|---|---|---|---|---|
| Gene (*P. aeruginosa*) | *amiE* | NA | Pseudomonas genome database: PA3366 | |
| Strain, strain background (*P. aeruginosa*) | PAO1 | PMID: 111024 | | |
| Strain, strain background (*P. aeruginosa*) | PAO1Δcrc | PMID: 20080802 | | |
| Strain, strain background (*Escherichia coli*) | XL1-Blue | Stratagene | | |
| Strain, strain background (*E. coli*) | BL21(DE3) | Novagen | | |
| Genetic reagent (*P. aeruginosa*) | *amiE*$_{6ARN}$ RNA | Microsynth, PMID: 29244160 | sequence: 5'-AAAAAUAACA ACAAGAAG-3' | |
| Antibody | anti-Crc (rabbit polyclonal) | Pineda | (1:5000) | |
| Antibody | anti-Hfq (rabbit polyclonal) | Pineda | (1:10000) | |
| Antibody | anti-S1 (rabbit polyclonal) | Pineda | (1:10000) | |
| Antibody | anti-rabbit IgG (goat polyclonal) | Sigma | Sigma: A3687 | conjugated with alkaline phosphatase, (1:10000) |
| Recombinant DNA reagent | pME9655 (plasmid) | PMID: 20080802 | | |
| Recombinant DNA reagent | pETM14lic-His$_6$Crc (plasmid) | PMID: 23717639 | | |
| Recombinant DNA reagent | pME4510crc$_{Flag}$ (plasmid) | PMID: 29244160 | | |
| Peptide, recombinant protein | Hfq | PMID: 21330354 | | |
| Peptide, recombinant protein | Crc | PMID: 23717639 | | |
| Software, algorithm | Relion3 | DOI: 10.7554/eLife.42166 | | |
| Software, algorithm | LocScale | DOI: 10.7554/eLife.27131 | | |
| Software, algorithm | LocalDeblur | DOI: 10.1101/433284 | | |

## Protein synthesis, purification and complex formation

*P. aeruginosa* Hfq and Crc were produced in *E. coli* and purified as described by *Sonnleitner et al. (2018)*. The synthetic 18-mer *amiE*$_{6ARN}$ RNA (5′-AAAAAUAACAACAAGAGG-3′) used in these studies consists of six tripartite binding motifs (*Sonnleitner and Bläsi, 2014*). The Hfq/Crc/RNA complex was prepared by first heating the *amiE*$_{6ARN}$ RNA at 95°C for 5 min followed by 50°C for 10 min and 37°C for 10 min. The RNA was then incubated with the Hfq hexamer at a 1:1 molar ratio on ice for 20 min to form a binary complex, then an equal molar ratio of Crc was added as recently described (*Sonnleitner et al., 2018*). The mixture was incubated on ice for 30 min prior to fractionation by size exclusion chromatography using a Superdex 200 column equilibrated in running buffer composed of 20 mM HEPES, pH 7.9, 10 mM KCl, 40 mM NaCl, 1 mM MgCl$_2$, and 2 mM TCEP (tris(2-carboxyethyl) phosphine). The peak fractions were buffer exchanged into 20 mM HEPES, pH 7.9, 10 mM KCl, 40 mM NaCl, 5 mM MgCl$_2$. Samples used for cross-linking were incubated with bis(sulfosuccinimidyl) suberate (BS$^3$) at 150 µM for 30 min on ice, followed by quenching at 37.5 mM Tris-HCl pH 8.0.

## CryoEM specimen preparation and data acquisition

Graphene oxide grids are prepared as described by *Pantelic et al. (2010)*. Briefly, 2 mg/ml of graphene oxide solution in water (Aldrich) was diluted ten times in water. After removing aggregation by spinning for 30 s at 300 rcf, 2 µl of graphene oxide solution was loaded on freshly glow discharged quantifoil Au-grids (R1.2/1.3, 300 mesh). Glow discharge was performed prior to graphene oxide coating at 45 mA for 60 s with an Edward Sputter Coater S150B at 0.2 m Bar at 0.75 KV. After the graphene oxide had been adsorbed for 1 min, the grids were washed 3 times with 20 µl water, then air-dried for 1 hr at room temperature prior to sample application. Specimens for cryoEM analysis were prepared by applying 2 µl of a 0.65 µM solution of the Hfq/Crc/RNA complex to the Quantifoil Au grids freshly coated with graphene oxide. After an adsorption time of 60 s, the grids were blotted for 10 s at a blot force of 5, then plunge frozen into liquid ethane using a Vitrobot (FEI). Images were recorded on a Krios G2, Falcon III direct electron detector at 300 kV operating in counting mode (*Supplementary file 1C*).

## Movie processing, single particle analysis, 3D reconstruction and refinement

Whole frame motion correction was performed on movies with motioncorr2 with dose weighting followed by CTF estimation using gctf (*Zhang, 2016*; *Zheng et al., 2017*). RELION-3.0 was used for data processing (*Scheres, 2012*). Final resolution estimates were calculated after the application of a soft binary mask and phase randomisation and determined based on the gold standard FSC = 0.143 criterion (*Scheres and Chen, 2012*; *Chen et al., 2013*).

For the $BS^3$ treated complex, after manually picking 3159 particles and using suitable references for autopicking, 482426 particles were used for early classifications. After three rounds of rejecting particles by 2D classification, 215774 particles were used for initial model generation and 3D classification. An initial model was generated using an SGD algorithm based on a small subset of particles with diverse orientations (*Punjani et al., 2017*). During 3D classification, three different complexes were resolved after 25 iterations with an angular sampling of 7.5°: 2Hfq:2Crc:2$amiE_{6ARN}$ (2:2:2), 2Hfq:3Crc:2$amiE_{6ARN}$ (2:3:2) and 2Hfq:4Crc:2$amiE_{6ARN}$ (2:4:2). To properly separate, validate and refine the three classes, the same 3D classification was rerun with the new 2:3:2 model as reference model, lowpass filtered to 20 Å resolution. C2 symmetry was observed and imposed for the 2:2:2 and 2:4:2 complexes. Each of the classes was then refined to sub-3.5 Å resolution, followed by per-particle frame alignment for movement correction and per-frame damage weighting. The resulting 'polished' particles were subjected to a final refinement round with solvent flattening. All reference models were lowpass filtered to 60 Å prior to refinement. The dominant class (2:2:2) had a resolution of 3.13 Å. Local resolution calculations were done with the RELION local resolution estimator, *Supplementary file 1A* (*Figure 1—figure supplement 1*). The 2:4:2 and 2:3:2 maps were sharpened locally with LocScale and LocalDeblur to better resolve the additional Crc components (*Jakobi et al., 2017*; *Ramírez-aportela et al., 2018*).

Crystal structures for *P. aeruginosa* Crc (PDB code 1U1S) and Hfq (PDB code 4JG3) were manually docked into the EM density map as rigid bodies in Chimera (*Pettersen et al., 2004*). The RNA 18-mers were manually built into the density using Coot (*Emsley et al., 2010*). Refmac5 and Phenix real-space refinement with global energy minimization, NCS-restraints, group B-factor and geometry restraints were used to iteratively refine the multi-subunit complexes at high resolution, followed by manual corrections for Ramachandran and geometric outliers in Coot and ISOLDE (*Supplementary file 1A*) (*Emsley et al., 2010*; *Murshudov et al., 2011*; *Afonine et al., 2012*; *Croll, 2018*). Model quality was evaluated with Procheck in CCP4 and MolProbity (*Williams et al., 2018*). In silico 2 Å maps were generated from the atomic models and FSC validation against the experimental maps was performed with the EMDB Fourier shell correlation server (EMBL-EBI) (*Figure 1—figure supplement 2B*).

## Bacterial strains and plasmids

The strains, plasmids and oligonucleotides used in this study are listed in *Supplementary file 1B and 1C*.

## Construction of plasmids encoding Crc variant proteins for in vivo translational repression assay

To test the proficiency of Crc mutant proteins to co-repress translation of a translational *amiE:lacZ* reporter gene, derivatives of plasmid pME4510crc$_{Flag}$ (*Supplementary file 1B*) were constructed by means of Quick change site directed mutagenesis (Agilent Technologies). Plasmid pME4510crc$_{Flag}$ was used together with the corresponding mutagenic oligonucleotide pairs (*Supplementary file 1C*). The parental plasmid templates were digested with *DpnI* and the mutated nicked circular strands were transformed into *E. coli* XL1-Blue, generating plasmids pME4510crc$_{(R140E)Flag}$, pME4510crc$_{(E142R)Flag}$, pME4510crc$_{(R229E)Flag}$, pME4510crc$_{(E193R)Flag}$, pME4510crc$_{(R230E)Flag}$, pME4510crc$_{(E142R,R229E)Flag}$, pME4510crc$_{(E193R,R230E)Flag}$, pME4510crc$_{(E193A,R230E)Flag}$, pME4510crc$_{(E142R,R230E)Flag}$ and pME4510crc$_{(E142R, R229E, R230E)Flag}$ and pME4510crc$_{(E142A, R229E, R230E)Flag}$.

## In vivo translational repression of an *amiE::lacZ* reporter gene in the presence of Crc variants

The ability of the Crc mutant proteins to repress translation of an *amiE::lacZ* reporter gene was tested in a PAO1 *crc* deletion strain bearing plasmids encoding the wt protein or the respective Crc variants (*Supplementary file 1B*) as described by *Sonnleitner et al. (2018)*. The β-galactosidase activities were determined as described (*Miller, 1972*). The β-galactosidase units in the different experiments were derived from two independent experiments.

## Construction of plasmids employed for the production of selected Crc mutant proteins

The R140E, E142R, R230E single aa exchanges in Crc were obtained by using the QuickChange site-directed mutagenesis protocol (Agilent Technologies). The plasmid pETM14lic-His$_6$Crc (*Supplementary file 1B*) was used together with the corresponding mutagenic oligonucleotide pairs (*Supplementary file 1C*). The entire plasmids were amplified with Pfu DNA polymerase (Thermo Scientific). The parental plasmid templates were digested with *DpnI* and the mutated nicked circular strands were transformed into *E. coli* XL1-Blue, generating plasmids pETM14lic-His6Crc$_{R140E}$, pETM14lic-His6Crc$_{E142R}$ and pETM14lic-His6Crc$_{R230E}$.

## Purification of Crc and Crc variants

The Crc protein and the Crc variants Crc$_{R140E}$, Crc$_{E142R}$ and Crc$_{R230E}$ were purified from *E. coli* strain BL21(DE3) harboring either plasmid pETM14lic-His Crc or the respective derivatives using Ni-affinity chromatography, followed by removal of the His$_6$-tag with GST-HRV14-3C ''PreScission'' protease as described by *Milojevic et al., 2013*.

## In vitro *co-IP studies*

The co-IP studies in the presence of 40 pmol of Hfq-hexamer, 120 pmol of Crc protein or of the respective Crc mutant proteins and 40 pmol *amiE$_{6ARN}$* RNA were performed as described (*Sonnleitner et al., 2018*).

## Western blot analyses

Equal amounts of proteins were separated on 12% SDS-polyacrylamide gels, and then electro-blotted onto a nitrocellulose membrane. The blots were blocked with 5% dry milk in TBS buffer, and probed with rabbit anti-Hfq (Pineda) and rabbit anti-Crc (Pineda) antibodies, respectively. Immunodetection of ribosomal protein S1 served as a loading control. The antibody-antigen complexes were visualized with alkaline-phosphatase conjugated secondary antibodies (Sigma) using the chromogenic substrates nitro blue tetrazolium chloride (NBT) and 5-Bromo-4-chloro-3-indolyl phosphate (BCIP).

## Acknowledgements

The coordinates and cryoEM maps have been deposited in the PDB and the EMBD. BFL, XYP and TD are supported by the Welcome Trust (200873/Z/16/Z). TD is also supported by an AstraZeneca

Studentship. UB and ES are supported by the Austrian Science Fund (FWF) (www.fwf.ac.at/en) [P28711-B22]. We thank our colleagues Jamie Blaza, Dima Chirgadze, Jiri Sponer, Miroslav Kreply, Kasia Bandyra, Steven Hardwick, Sjors Scheres, Joerg Vogel, Armin Resch and Nguyen Thi Bach Hue for advice, helpful discussions and support. For access and help at facilities, we thank Giuseppe Cannon and staff at the MRC-LMB EM Facility and Kasim Sader at Thermo Fisher Scientific Pharma Cryo-oEM Facility, Nanoscience Centre of University of Cambridge.

## Additional information

### Funding

| Funder | Grant reference number | Author |
|---|---|---|
| Wellcome Trust | 200873/Z/16/Z | Ben F Luisi<br>Xue Yuan Pei<br>Tom Dendooven |
| AstraZeneca | Studentship | Tom Dendooven |
| Austrian Science Fund | P28711-B22 | Udo Bläsi<br>Elisabeth Sonnleitner |

The funders had no role in study design, data collection and interpretation, or the decision to submit the work for publication.

### Author contributions
Xue Yuan Pei, Data curation, Formal analysis, Investigation; Tom Dendooven, Elisabeth Sonnleitner, Conceptualization, Data curation, Formal analysis, Investigation, Methodology; Shaoxia Chen, Data curation, Formal analysis; Udo Bläsi, Conceptualization, Formal analysis, Funding acquisition, Investigation, Project administration; Ben F Luisi, Conceptualization, Funding acquisition, Investigation, Project administration

### Author ORCIDs
Ben F Luisi (iD) http://orcid.org/0000-0003-1144-9877

### Decision letter and Author response
Decision letter https://doi.org/10.7554/eLife.43158.028
Author response https://doi.org/10.7554/eLife.43158.029

## Additional files

### Supplementary files
• Supplementary file 1. Resolution of maps and angular distributions of 2D images. (**A**) Summary of the data collection parameters and the reconstruction and refinement statistics for the Hfq:Crc:RNA complexes. (**B**) List of the plasmids and strains used in this study. (**C**) Oligonucleotides used in this study.
DOI: https://doi.org/10.7554/eLife.43158.013
• Transparent reporting form
DOI: https://doi.org/10.7554/eLife.43158.014

### Data availability
CryoEM data have been deposited in the Electron Microscopy Data Bank.

The following datasets were generated:

| Author(s) | Year | Dataset title | Dataset URL | Database and Identifier |
|---|---|---|---|---|
| Tom Dendooven, Xue Yuan Pei, Elisabeth Sonnleitner, | 2019 | Hfq/Crc/RNA cpmplex 2:4:2 ratio | https://www.ebi.ac.uk/pdbe/entry/pdb/6o1m | Protein Data Bank in Europe, 6o1m |

| | | | | |
|---|---|---|---|---|
| Shaoxia Chen, Udo Blasi, Ben F Luisi | | | | |
| Tom Dendooven, Xue Yuan Pei, Elisabeth Sonnleitner, Shaoxia Chen, Udo Blasi, Ben F Luisi | 2019 | Hfq/Crc/RNA complex 2:2:2 ratio | http://www.ebi.ac.uk/pdbe/entry/emdb/EMD-0590 | Electron Microscopy Data Bank, EMD-0590 |
| Tom Dendooven, Xue Yuan Pei, Elisabeth Sonnleitner, Shaoxia Chen, Udo Blasi, Ben F Luisi | 2019 | Hfq/Crc/RNA complex 2:3:2 ratio | http://www.ebi.ac.uk/pdbe/entry/emdb/EMD-0591 | Electron Microscopy Data Bank, EMD-0591 |
| Tom Dendooven, Xue Yuan Pei, Elisabeth Sonnleitner, Shaoxia Chen, Udo Blasi, Ben F Luisi | 2019 | Hfq/Crc/RNA complex 2:4:2 ratio | http://www.ebi.ac.uk/pdbe/entry/emdb/EMD-0592 | Electron Microscopy Data Bank, EMD-0592 |
| Tom Dendooven, Xue Yuan Pei, Elisabeth Sonnleitner, Shaoxia Chen, Udo Bläsi, Ben F Luisi | 2019 | Hfq/Crc/RNA complex 2:2:2 ratio | https://www.ebi.ac.uk/pdbe/entry/pdb/6o1k | Protein Data Bank in Europe, 6o1k |
| Tom Dendooven, Xue Yuan Pei, Elisabeth Sonnleitner, Shaoxia Chen, Udo Bläsi | 2019 | Hfq/Crc/RNA complex 2:3:2 ratio | https://www.ebi.ac.uk/pdbe/entry/pdb/6o1l | Protein Data Bank in Europe, 6o1l |

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
