## [Decision Letter]

Thank you for submitting your article "Architectural principles for Hfq/Crc-mediated regulation of gene expression" for consideration by *eLife*. Your article has been reviewed by three peer reviewers, one of whom is a member of our Board of Reviewing Editors, and the evaluation has been overseen by John Kuriyan as the Senior Editor. The reviewers have opted to remain anonymous.

The reviewers have discussed the reviews with one another and the Reviewing Editor has drafted this decision to help you prepare a revised submission.

Summary:

The authors present an exceptionally significant set of structures that reveal how Hfq and associated cofactor proteins interact with mRNA in order to facilitate translation and coordinate interactions with sRNA molecules. This is really the first glimpse of these important complexes and the molecular recognition and assembly pathways that are in play. It is truly exceptional work and the reviewers were very enthusiastic. That said, there is a need to restructure the paper so that it is focused on the most significant issues, and to provide stronger functional data to support the structural findings, as described below and in the reviews.

Essential revisions:

1) In the reviews and in subsequent conversation among the reviewers, there was unanimity on the need to restructure the paper so that it is focused on the main complexes and their significance to biology. The paper should be organized in a more effective way (please refer to the comments) and that major findings should be presented and discussed in light of their importance and relevance to the field at large. The text needs to be rewritten for clarity and context.

2) There was also general agreement that the supporting biochemical data should be strengthened, and the experiments suggested in the major comments by the referees should be conducted in some form in order to support the work. While the reviewers do not want to place undue burden on the investigators, there are places in the manuscript where critical supporting functional experiments still need to be included, as indicated. In addition, more attention must be devoted to a description of the methodological approaches, as these impact the findings directly in this case.

*Reviewer #1:*

This is a fascinating manuscript that not only provides new insights into the molecular mechanism of Hfq function and its interplay with cofactors, but also contains lessons for understanding regulatory RNA-protein complexes in general. The structures themselves are intriguing and they provide a trove of information on molecular recognition strategies within RNA-protein complexes. For example, Figure 2 is particularly interesting as no single type of interaction is completely new, but the way they are used on concert is extremely informative. The one weakness of the paper (in addition to a relatively poor presentation of the different types of complexes) is the weak supporting data on the protein-protein interactions. Attention to the points below would strengthen the paper and enhance its impact.

1) Subsection “Protein synthesis, purification and complex formation”: More experimental details about complex assembly are useful but also important for mechanistic understanding. Was it necessary to incubate the RNA and the Hfq hexamer first before adding Crc? What order of addition experiments were done to arrive at this procedure? How did the authors determine the stoichiometry for Crc? Combining the RNA/Hfq/Crc in a 1:1:1 ratio is hard to understand given the ultimate stoichiometries observed in the particles – so why didn't the authors go back and try again? This is important, as it is germane to the biological relevance described later in the paper. Given the distribution of particle stoichiometries (Figure 1), why were other assembly conditions not tried and if they were, why were they not used? Some of this is discussed in Sonnleitner, and specific results from that paper could be cited more clearly in support of the present manuscript.

2) Subsection “An ensemble of Hfq/Crc/*amiE*_6ARN_RNA assemblies”, first paragraph: Mild crosslinking conditions were purportedly used. Were others tried? The end of the paragraph says crosslinking does not affect structure. So why was it used? What did it help?

3) Subsection “An ensemble of Hfq/Crc/*amiE*_6ARN_RNA assemblies”, last paragraph: In the first part of the paper, there is too little focus on the overall organization and features of the 2:2:2 complex and far too much focus on the organization of complexes in the different stoichiometries – it distracts from the basic elements that are common to all three and which are likely to be of central importance. For example, the discussion of symmetry is interesting, but not central and will not help with general understanding of function by most readers. Instead, in this part of the paper, it would be good to focus on overall organization of the 2:2:2 given that it is later suggested to be the most relevant one. In the opinion of this reviewer, the first part of the paper should not have figures and text with such an emphasis on alternative stoichiometries.

4) The pull-down experimental data shown in Figure 2—figure supplement 1 are qualitative and not well explained. It's not clear to the reader how the raw data demonstrate complex formation or specific interfaces unambiguously, at least as shown. Given the affinities known for these proteins, using EMSA or size/exclusion+MALS as experimental techniques on mutants would significantly strengthen this paper. That said, a clear and compelling explanation of why the existing data are sufficient might obviate the need for these experiments.

5) Subsection “Function, origins and validation of subunit cooperativity in the 2:2:2 complex”, third paragraph. The authors state that mutant Crc_R140E_ did not co-IP with Hfq – but the data in the Figure 2—figure supplement 1B look just like WT Crc, so this seems to undermine the authors' point, unless there is some type of misunderstanding. Another issue: it appears from the structure that R140 also participates in the Crc1-Crc2 interface by contacting E193 (in the 4jg3 crystal structure of Crc), whereas the authors only discuss its role in RNA binding. E193, instead is described by the authors to be relevant only in the Crc3-Crc4 interface. A more balanced and careful discussion of these interfaces is therefore suggested.

6) Subsection “A specialised and recurring RNA conformation in Hfq-mediated regulation”, last paragraph: The molecular determinants for A and R specificity are among the more central issues for this field, but this is not covered extensively. H-bonds consistent with a specificity pocket are observed and noted (Figure 4), but these do not necessarily confer specificity. Have the amino acids interacting with the N6 amine and N1 been mutated and tested previously and is this why the authors have not made these mutations? If so, the results should be discussed. That said, it would seem reasonable to test the effect of these mutants in the context of this more biologically relevant RNA sequence.

7) Discussion, third paragraph: It would be best if the stepwise assembly process of the higher-order complex were tested experimentally through order of additional experiments, SEC-mals or similarly informative biophysical experiments given the emphasis on the 2:4:2 complex given within the paper. This gets back to point #1, above. Otherwise, the authors need to change the speculative language in the text.

*Reviewer #2:*

This work presents exciting high resolution cryo EM structures (and some functional validation experiments) showing how the Crc 'helper' protein recognizes a composite Hfq-RNA surface and thus can act as a cofactor of Hfq-mediated RNA regulation in bacteria. The structure represents an important advance in our molecular understanding of Hfq function and demonstrates that it should be feasible to analyze similar Hfq complexes by cryo EM in the future. I'd enthusiastically support publication of the work in *eLife* after the overall description of the structure is improved to be more amenable to more generally interested readers.

The structure is not very clearly described and structure description is difficult to follow even for readers who have previously seen similar structures. A more streamlined description of the overall structure going from known interactions (Hfq-RNA) to the novel and exciting features (role of Crc in complex assembly), and a precise assignment of interface names (without swapping between symmetry-related interfaces) would help the general reader to better appreciate the principles of complex assembly and RNA recognition in the present case. Otherwise the most exciting insight from this work, i.e. how Crc improves RNA recognition and stabilizes the Hfq-RNA interaction gets lost too much in structural details (see point 2).

The Introduction could profit from some clarifications to help general readers understand and evaluate the significance of the present work (see point 1).

Some of the references to data by Sonnleitner et al., 2018, are rather vague and corresponding statements remain obscure without detailed prior knowledge of the work by Sonnleitner (see point 3).

1) Introduction

1.1) Clarify early in the general introduction of Hfq that Hfq controls mRNA translation not only indirectly (via small RNAs) but also directly (i.e. independently of small RNAs) by binding A-rich sequences at/near the mRNA translation initiation sites. Later on in the Introduction, mention that *amiE* RNA is such a case where Hfq's distal side binds A-rich sequences. This would also be a good occasion to mention that A-rich RNA binding to Hfq is principally understood in structural terms (Link et al.), but that the role of Crc in this context remained obscure and is the topic of the present study.

1.2) Clarify in the Introduction that Crc is a rather specialized protein that is not generally found in all bacteria but rather in a limited subset including *Pseudomonas* but excluding e.g. *Enterobacteriaceae* such as *Escherichia coli*. Such a statement does not limit the scope of the paper but helps the reader understand why the presented structure is so useful also for the design of selective drugs/inhibitors.

1.3) Mention in the Introduction that Crc is a very interesting member of the EEP protein family (Exonuclease/Endonuclease/Phosphatase) although key active site residues in the active site cleft are mutated. Information of the orientation of the active site cleft in the complex may be helpful in overall structure description (facing outward, laterally). Although Crc is catalytically inactive, I wonder whether it might bind phosphorylated sugars or metabolites in its active site cleft and hence helps to sense available carbon sources. Is anything known in this direction?

2) Structure description

2.1) Most importantly, I would suggest to improve and reorganize the description of the overall complex and of the individual interfaces. The authors use a defined nucleotide sequence (5'-AAA-AAU-AAC-AAC-AAG-AAG-3', note it down at the beginning of the Results section), and they assign nucleotide numbers in their structure model. In contrast to Link et al., the RNA density is hence not averaged over the Hfq hexamer and the 5' and 3' ends are visible. Consequently, it should be possible to number Hfq monomers from one (nucleotides 1-3) to six (nucleotides 16-18) and use these Hfq monomer numbers in the description of the various interfaces.

2.2) It seems more 'intuitive' to begin complex description with the Hfq-RNA complex and to point out what is novel/unique/different as compared to previous such complexes published by Link et al., 2009, or also Horstmann et al., 2012. In the following, all of the Crc interactions (i.e. with Crc1, Crc2 and Crc3) can then be described with only one of the Hfq-RNA complexes (since all of the interactions with the second Hfq ring are symmetry-related).

2.3) In a second step, one then could describe how the twofold symmetric Crc1-Crc2 dimer recognizes and binds over the Hfq-RNA interface, how the Hfq-RNA complex stabilizes the Crc1-Crc2 dimer and why the 2:2:2 complex forms cooperatively in an all-or-nothing fashion. Here one can then also describe and validate the Crc1-Crc2 interface in comparison to existing crystal packing interactions in Milojevic et al., 2013, (but also previously in Wei et al., 2012).

2.4) In a third step, one would then describe how Crc3 recognizes the Hfq-RNA complex, maybe in comparison to Crc1 and Crc2. Does Crc3 interact at all with Crc1 or Crc2? Finally, one could compare the Crc3-Crc4 interface to the Crc1-Crc2 interface and describe how the two Crc dimers are oriented with respect to each other in the full complex. As the authors point out, the Crc3-Crc4 interface is somewhat less important than the Crc1-Crc2 interface, because Crc3 and Crc4 can be added in a stepwise fashion.

3) Structure validation

3.1) Regarding structure validation, the authors mainly test for the different Crc dimer interfaces and only using duplicates in the translational repression reporter assay. It might be worthwhile to solidify these results and also test E142A and E193A mutants in order to see whether charge compensation really is the correct interpretation for the success of the rescue experiments. It also seems from the Crc crystal structure that E193 participates as well in the Crc1-Crc2 dimer interface. Maybe tone done a bit the exclusive importance of E193 for the Crc3-Crc4 interface. Moreover, since the authors have access to size exclusion chromatography and MALS instrumentation, they could for example test as well to which degree changes in the RNA sequence affect recognition by Crc and complex formation. Although not strictly required, these experiments clearly would strengthen the paper.

3.2) On various occasions, the authors rather vaguely point to previous results obtained by Sonnleitner et al., 2018, in support of the present cryo-EM structure. It would be good to also cite numeric values (e.g. subsection “An ensemble of Hfq/Crc/*amiE*_6ARN_RNA assemblies”, first paragraph, ~220 kDa?) or define more precisely which mutants, experiments or data they have in mind when citing this paper (e.g. subsection “Function, origins and validation of subunit cooperativity in the 2:2:2 complex”, third paragraph).

3.3) Finally, in the Discussion, the authors might want to stress a bit more that this is the first time that structural information was obtained on how a 'helper' protein of Hfq can assist and specify the role of Hfq in direct translational repression. They also might want to explain a bit more extensively and speculate on what is the added value of Crc in the complex as compared to a simple recognition of the A-rich sequence by Hfq. In the second paragraph of the Discussion, it would again be important to be more precise and distinguish between thermodynamic and kinetic stability – otherwise this passage of the text is quite confusing.

*Reviewer #3:*

This study reveals for the first time the molecular mechanism by which two major regulatory proteins Hfq and Crc cooperate to regulate the translation of mRNAs in *Pseudomonas aeruginosa*. High-resolution structures of complexes formed by Hfq, Crc and the operator regulatory region of *amiE* mRNA were solved using cryo-EM. Based on these structures, specific mutations were introduced into Crc to analyze their ability to form RNP and to regulate *amiE* translation. Taken together, the authors visualized how Crc and Hfq assemble around a specific binding site on *amiE* mRNA to regulate translation initiation. This work is incredible and illustrates the importance of RNP assembly in the regulation of gene expression in bacteria. This work beautifully showed how a strong RNA binder (Hfq) is able to recruit proteins that are not able solely to bind RNA (Crc) in order to expand the regulatory networks. This work is incredible and opens up new ideas, i.e. to search for novel translational regulators, but also should pave the way to identify novel strategies interfering with the metabolism of pathogenic bacteria. This is an important paper, which also reconciles previous published works on Crc function.

---

## [Author Response]

Reviewer #1:

This is a fascinating manuscript that not only provides new insights into the molecular mechanism of Hfq function and its interplay with cofactors, but also contains lessons for understanding regulatory RNA-protein complexes in general. The structures themselves are intriguing and they provide a trove of information on molecular recognition strategies within RNA-protein complexes. For example, Figure 2 is particularly interesting as no single type of interaction is completely new, but the way they are used on concert is extremely informative. The one weakness of the paper (in addition to a relatively poor presentation of the different types of complexes) is the weak supporting data on the protein-protein interactions. Attention to the points below would strengthen the paper and enhance its impact.1)Subsection “Protein synthesis, purification and complex formation”: More experimental details about complex assembly are useful but also important for mechanistic understanding. Was it necessary to incubate the RNA and the Hfq hexamer first before adding Crc? What order of addition experiments were done to arrive at this procedure? How did the authors determine the stoichiometry for Crc? Combining the RNA/Hfq/Crc in a 1:1:1 ratio is hard to understand given the ultimate stoichiometries observed in the particles – so why didn't the authors go back and try again? This is important, as it is germane to the biological relevance described later in the paper. Given the distribution of particle stoichiometries (Figure 1), why were other assembly conditions not tried and if they were, why were they not used? Some of this is discussed in Sonnleitner, and specific results from that paper could be cited more clearly in support of the present manuscript.

We have not exhaustively explored the order of addition to optimise the reconstitution, but we reasoned that the Hfq/RNA interaction is the stronger interaction and would nucleate the assembly. The conditions and search for optimal conditions were described in Sonnleitner et al., 2018, and we have added a comment to the revised text (first paragraph: Materials and methods). We also mentioned that SEC-MALS (Sonnleitner et al., 2018) have excluded a simple 1:1:1 assembly (subsection “An ensemble of Hfq/Crc/*amiE*_6ARN_RNA assemblies”).

2) Subsection “An ensemble of Hfq/Crc/amiE_6ARN_RNA assemblies”, first paragraph: Mild crosslinking conditions were purportedly used. Were others tried? The end of the paragraph says crosslinking does not affect structure. So why was it used? What did it help?

We used the bifunctional crosslinker BS3 to prepare specimens for all high-resolution datasets to maintain the complexes during grid preparations. To ensure that the observed assemblies were not artefacts from the crosslinking procedure, we subsequently collected a smaller data set without crosslinker and found that the structure remained unchanged within the resolution limits of the data for the models being compared. This is now explicitly mentioned in the subsection “An ensemble of Hfq/Crc/*amiE*_6ARN_RNA assemblies”, and the results are presented in Supplementary file 1, Table S1. Given that no crosslinking artefacts were found, we did not further use/optimise the non-crosslinked datasets.

3) Subsection “An ensemble of Hfq/Crc/amiE_6ARN_RNA assemblies”, last paragraph: In the first part of the paper, there is too little focus on the overall organization and features of the 2:2:2 complex and far too much focus on the organization of complexes in the different stoichiometries – it distracts from the basic elements that are common to all three and which are likely to be of central importance. For example, the discussion of symmetry is interesting, but not central and will not help with general understanding of function by most readers. Instead, in this part of the paper, it would be good to focus on overall organization of the 2:2:2 given that it is later suggested to be the most relevant one. In the opinion of this reviewer, the first part of the paper should not have figures and text with such an emphasis on alternative stoichiometries.

We have re-organised the text to focus first on the 2:2:2 complex, and then turn to the other complexes.

4) The pull-down experimental data shown in Figure 2—figure supplement 1 are qualitative and not well explained. It's not clear to the reader how the raw data demonstrate complex formation or specific interfaces unambiguously, at least as shown. Given the affinities known for these proteins, using EMSA or size/exclusion+MALS as experimental techniques on mutants would significantly strengthen this paper. That said, a clear and compelling explanation of why the existing data are sufficient might obviate the need for these experiments.The IP data in Figure 2—figure supplement 1A are qualitative but suggest that there is a good agreement between the in vivo data and the in vitro results, indicating that the purified components are sufficient to recapitulate the complexity of the in vivo process. The method for the pull down is described in Sonnleitner et al., 2018, and we have changed the text to explain the data more clearly (subsection “Function, origins and validation of subunit cooperativity in the 2:2:2 complex”, second paragraph).5) Subsection “Function, origins and validation of subunit cooperativity in the 2:2:2 complex”, third paragraph. The authors state that mutant Crc_R140E_ did not co-IP with Hfq – but the data in the Figure 2—figure supplement 1B look just like WT Crc, so this seems to undermine the authors' point, unless there is some type of misunderstanding. Another issue: it appears from the structure that R140 also participates in the Crc1-Crc2 interface by contacting E193 (in the 4jg3 crystal structure of Crc), whereas the authors only discuss its role in RNA binding. E193, instead is described by the authors to be relevant only in the Crc3-Crc4 interface. A more balanced and careful discussion of these interfaces is therefore suggested.

The data in Figure 2—figure supplement 1B show the input levels for the proteins used in the pulldowns for Figure 2—figure supplement 1A. In the 4jg3 crystal structure R140 indeed participates in the Crc1-Crc2 dimer interface, however, in our maps the density for R140 clearly shows a rotamer shift pulling the R140 side chain away from E193 and towards A3 of *amiE* in the Entry/Exit site. This has now been addressed in the revised text. E193 does participate in the dimer interface of Crc3-Crc4 via contacts with R230 and R233 (Figure 4A and 4C), as discussed in the text.

6) Subsection “A specialised and recurring RNA conformation in Hfq-mediated regulation”, last paragraph: The molecular determinants for A and R specificity are among the more central issues for this field, but this is not covered extensively. H-bonds consistent with a specificity pocket are observed and noted (Figure 4), but these do not necessarily confer specificity. Have the amino acids interacting with the N6 amine and N1 been mutated and tested previously and is this why the authors have not made these mutations? If so, the results should be discussed. That said, it would seem reasonable to test the effect of these mutants in the context of this more biologically relevant RNA sequence.

The network of hydrogen bonding interactions with the A and R in the A-R-N pattern in *Pseudomonas* aeruginosa Hfq are consistent with the interactions observed by Link et al., 2009, for the *E. coli* Hfq with polyA_18_. The interactions do specify sequence preferences in a way that is consistent with binding affinities and rationalisation from the structure. For instance, the interactions with N1 (by the carbonyl of Q33) and N7 (by the amide of Q33) specify A as they are not compatible with a G, which form a repulsive contact of the O6 and peptide carbonyl. These interactions cannot be tested by mutagenesis because they do not involve side chain changes. Other changes in Hfq cannot be tested easily in vivo because of the pleotropic effect on many other RNA processes mediated by Hfq. We have changed the text to discuss the basis for the specificity.

We would also like to mention that our colleagues Jiri Sponer and Miroslav Krepl have found that the A in the A-R-N can switch between syn and anti-conformations in molecular dynamics simulations, and that both conformations are only compatible with A.

7) Discussion, third paragraph: It would be best if the stepwise assembly process of the higher-order complex were tested experimentally through order of additional experiments, SEC-mals or similarly informative biophysical experiments given the emphasis on the 2:4:2 complex given within the paper. This gets back to point #1, above. Otherwise, the authors need to change the speculative language in the text.

The proposed assembly process is inferred from the fact that the assemblies all share the same core 2:2:2 complex. Nevertheless, we have changed the text as suggested so that it is less speculative. The 2:4:2 complex is demonstrated to be important in vivo, as shown by the second site revertants (Figure 4B).

Reviewer #2:

This work presents exciting high resolution cryo EM structures (and some functional validation experiments) showing how the Crc 'helper' protein recognizes a composite Hfq-RNA surface and thus can act as a cofactor of Hfq-mediated RNA regulation in bacteria. The structure represents an important advance in our molecular understanding of Hfq function and demonstrates that it should be feasible to analyze similar Hfq complexes by cryo EM in the future. I'd enthusiastically support publication of the work in eLife after the overall description of the structure is improved to be more amenable to more generally interested readers.The structure is not very clearly described and structure description is difficult to follow even for readers who have previously seen similar structures. A more streamlined description of the overall structure going from known interactions (Hfq-RNA) to the novel and exciting features (role of Crc in complex assembly), and a precise assignment of interface names (without swapping between symmetry-related interfaces) would help the general reader to better appreciate the principles of complex assembly and RNA recognition in the present case. Otherwise the most exciting insight from this work, i.e. how Crc improves RNA recognition and stabilizes the Hfq-RNA interaction gets lost too much in structural details (see point 2).The Introduction could profit from some clarifications to help general readers understand and evaluate the significance of the present work (see point 1).Some of the references to data by Sonnleitner et al., 2018, are rather vague and corresponding statements remain obscure without detailed prior knowledge of the work by Sonnleitner (see point 3).1) Introduction1.1) Clarify early in the general introduction of Hfq that Hfq controls mRNA translation not only indirectly (via small RNAs) but also directly (i.e. independently of small RNAs) by binding A-rich sequences at/near the mRNA translation initiation sites. Later on in the Introduction, mention that amiE RNA is such a case where Hfq's distal side binds A-rich sequences. This would also be a good occasion to mention that A-rich RNA binding to Hfq is principally understood in structural terms (Link et al.), but that the role of Crc in this context remained obscure and is the topic of the present study.

We have changed the “Introduction” as suggested.

1.2) Clarify in the Introduction that Crc is a rather specialized protein that is not generally found in all bacteria but rather in a limited subset including Pseudomonas but excluding e.g. Enterobacteriaceae such as Escherichia coli. Such a statement does not limit the scope of the paper but helps the reader understand why the presented structure is so useful also for the design of selective drugs/inhibitors.

We have made the suggested changes.

1.3) Mention in the Introduction that Crc is a very interesting member of the EEP protein family (Exonuclease/Endonuclease/Phosphatase) although key active site residues in the active site cleft are mutated. Information of the orientation of the active site cleft in the complex may be helpful in overall structure description (facing outward, laterally). Although Crc is catalytically inactive, I wonder whether it might bind phosphorylated sugars or metabolites in its active site cleft and hence helps to sense available carbon sources. Is anything known in this direction?

We have included the suggested changes in the text. The suggestion that Crc might bind metabolites is an intriguing concept, but we are not aware of any evidence for this.

2) Structure description2.1) Most importantly, I would suggest to improve and reorganize the description of the overall complex and of the individual interfaces. The authors use a defined nucleotide sequence (5'-AAA-AAU-AAC-AAC-AAG-AAG-3', note it down at the beginning of the Results section), and they assign nucleotide numbers in their structure model. In contrast to Link et al., the RNA density is hence not averaged over the Hfq hexamer and the 5' and 3' ends are visible. Consequently, it should be possible to number Hfq monomers from one (nucleotides 1-3) to six (nucleotides 16-18) and use these Hfq monomer numbers in the description of the various interfaces.

The proposed assignment numbers to Hfq monomers and the corresponding A-R-N nucleotides might help to describe each ARN-interface independently, but we feel that this would be unnecessary because they are all very similar.

2.2) It seems more 'intuitive' to begin complex description with the Hfq-RNA complex and to point out what is novel/unique/different as compared to previous such complexes published by Link et al., 2009, or also Horstmann et al., 2012. In the following, all of the Crc interactions (i.e. with Crc1, Crc2 and Crc3) can then be described with only one of the Hfq-RNA complexes (since all of the interactions with the second Hfq ring are symmetry-related).

We have reorganised the text to begin the description of the Hfq-RNA interactions and included a section in the “Results” and in the “Discussion” to compare our structures to that of Link et al. and Horstman et al. structures (and reference to Horstmann et al., 2012). A supplementary figure to show this has been added as well (Figure 2—figure supplement 1).

2.3) In a second step, one then could describe how the twofold symmetric Crc1-Crc2 dimer recognizes and binds over the Hfq-RNA interface, how the Hfq-RNA complex stabilizes the Crc1-Crc2 dimer and why the 2:2:2 complex forms cooperatively in an all-or-nothing fashion. Here one can then also describe and validate the Crc1-Crc2 interface in comparison to existing crystal packing interactions in Milojevic et al., 2013, (but also previously in Wei et al., 2012).

We have made the suggested change to the text and included the reference Wei et al., 2012.

2.4) In a third step, one would then describe how Crc3 recognizes the Hfq-RNA complex, maybe in comparison to Crc1 and Crc2. Does Crc3 interact at all with Crc1 or Crc2? Finally, one could compare the Crc3-Crc4 interface to the Crc1-Crc2 interface and describe how the two Crc dimers are oriented with respect to each other in the full complex. As the authors point out, the Crc3-Crc4 interface is somewhat less important than the Crc1-Crc2 interface, because Crc3 and Crc4 can be added in a stepwise fashion.

We have made the suggested change to the text.

3) Structure validation3.1) Regarding structure validation, the authors mainly test for the different Crc dimer interfaces and only using duplicates in the translational repression reporter assay. It might be worthwhile to solidify these results and also test E142A and E193A mutants in order to see whether charge compensation really is the correct interpretation for the success of the rescue experiments. It also seems from the Crc crystal structure that E193 participates as well in the Crc1-Crc2 dimer interface. Maybe tone done a bit the exclusive importance of E193 for the Crc3-Crc4 interface. Moreover, since the authors have access to size exclusion chromatography and MALS instrumentation, they could for example test as well to which degree changes in the RNA sequence affect recognition by Crc and complex formation. Although not strictly required, these experiments clearly would strengthen the paper.

As suggested, we have introduced the E142A exchange in the R229/230E background and the E193A exchange in the R230E background. The triple substitution Crc_E142A,R229E,R230E_ has a similar in vivo activity as the Crc_E142R,R229E,R230E_ variant.

For the Crc_E193A,R230E_ variant, the repressive activity was also comparable to that of the Crc_E193R, R230_ mutant protein.

These results suggest that the rescue of the deleterious mutations does not require charge compensation, but only removing any potential charge repulsion. We have mentioned this in the revised text.

In the 4jg3 crystal structure R140 indeed participates in the Crc1-Crc2 dimer interface, however, in our maps the density for R140 clearly shows a rotamer shift pulling away the R140 sidechain from E193 and towards A3 from *amiE* in the Entry/Exit site. This has now been mentioned in the revised text. E193 does participate in the dimer interface of Crc3-Crc4 *via* contacts with R230 and R233 (Figure 4A and 4C), as discussed in the text.

3.2) On various occasions, the authors rather vaguely point to previous results obtained by Sonnleitner et al., 2018, in support of the present cryo-EM structure. It would be good to also cite numeric values (e.g. subsection “An ensemble of Hfq/Crc/amiE_6ARN_RNA assemblies”, first paragraph, ~220 kDa?) or define more precisely which mutants, experiments or data they have in mind when citing this paper (e.g. subsection “Function, origins and validation of subunit cooperativity in the 2:2:2 complex”, third paragraph).

We have clarified the text.

3.3) Finally, in the Discussion, the authors might want to stress a bit more that this is the first time that structural information was obtained on how a 'helper' protein of Hfq can assist and specify the role of Hfq in direct translational repression. They also might want to explain a bit more extensively and speculate on what is the added value of Crc in the complex as compared to a simple recognition of the A-rich sequence by Hfq. In the second paragraph of the Discussion, it would again be important to be more precise and distinguish between thermodynamic and kinetic stability – otherwise this passage of the text is quite confusing.

We have changed the text as suggested.